

# Translating genomics into practice for real-time surveillance and response to carbapenemase-producing Enterobacteriaceae: evidence from a complex multi-institutional KPC outbreak

Jason C. Kwong[1,2,3,4], Courtney R. Lane[4,5], Finn Romanes[5], Anders Gonçalves da Silva[1,4], Marion Easton[5], Katie Cronin[6], Mary Jo Waters[6], Takehiro Tomita[4], Kerrie Stevens[4], Mark B. Schultz[1,2,4], Sarah L. Baines[1,2], Norelle L. Sherry[1,2,3], Glen P. Carter[1,2], Andre Mu[1,2], Michelle Sait[4], Susan A. Ballard[1,4], Torsten Seemann[1,7], Timothy P. Stinear[1,2] and Benjamin P. Howden[1,2,3,4]

[1] Doherty Applied Microbial Genomics, The University of Melbourne at The Peter Doherty Institute for Infection and Immunity, Melbourne, VIC, Australia
[2] Department of Microbiology and Immunology, The University of Melbourne at The Peter Doherty Institute for Infection and Immunity, Melbourne, VIC, Australia
[3] Department of Infectious Diseases, Austin Health, Heidelberg, VIC, Australia
[4] Microbiological Diagnostic Unit Public Health Laboratory, The University of Melbourne at The Peter Doherty Institute for Infection and Immunity, Melbourne, VIC, Australia
[5] Health Protection Branch, Department of Health and Human Services, Victoria State Government, Melbourne, VIC, Australia
[6] Department of Microbiology, St Vincent's Hospital Melbourne, Fitzroy, VIC, Australia
[7] Melbourne Bioinformatics, The University of Melbourne, Carlton, VIC, Australia

Corresponding author
Benjamin P. Howden,
bhowden@unimelb.edu.au

## ABSTRACT

**Background:** Until recently, *Klebsiella pneumoniae* carbapenemase (KPC)-producing Enterobacteriaceae were rarely identified in Australia. Following an increase in the number of incident cases across the state of Victoria, we undertook a real-time combined genomic and epidemiological investigation. The scope of this study included identifying risk factors and routes of transmission, and investigating the utility of genomics to enhance traditional field epidemiology for informing management of established widespread outbreaks.

**Methods:** All KPC-producing Enterobacteriaceae isolates referred to the state reference laboratory from 2012 onwards were included. Whole-genome sequencing was performed in parallel with a detailed descriptive epidemiological investigation of each case, using Illumina sequencing on each isolate. This was complemented with PacBio long-read sequencing on selected isolates to establish high-quality reference sequences and interrogate characteristics of KPC-encoding plasmids.

**Results:** Initial investigations indicated that the outbreak was widespread, with 86 KPC-producing Enterobacteriaceae isolates (*K. pneumoniae* 92%) identified from 35 different locations across metropolitan and rural Victoria between 2012 and 2015. Initial combined analyses of the epidemiological and genomic data resolved the outbreak into distinct nosocomial transmission networks, and identified healthcare facilities at the epicentre of KPC transmission. New cases were assigned to transmission networks in real-time, allowing focussed infection control efforts.

PacBio sequencing confirmed a secondary transmission network arising from inter-species plasmid transmission. Insights from Bayesian transmission inference and analyses of within-host diversity informed the development of state-wide public health and infection control guidelines, including interventions such as an intensive approach to screening contacts following new case detection to minimise unrecognised colonisation.

**Conclusion:** A real-time combined epidemiological and genomic investigation proved critical to identifying and defining multiple transmission networks of KPC Enterobacteriaceae, while data from either investigation alone were inconclusive. The investigation was fundamental to informing infection control measures in real-time and the development of state-wide public health guidelines on carbapenemase-producing Enterobacteriaceae surveillance and management.

# INTRODUCTION

Carbapenemase-producing Enterobacteriaceae (CPE) are among the most urgent antimicrobial resistance threats worldwide (*Centers for Disease Control and Prevention, US Department of Health and Human Services, 2013*). In addition to producing the carbapenemase, capable of inactivating almost all beta-lactam antibiotics, including penicillins, cephalosporins and carbapenems, these organisms frequently harbour multiple other antibiotic resistance genes and mutations (*Queenan & Bush, 2007*). Few therapeutic options exist, and the available options are often limited by tolerability and efficacy (*Falagas et al., 2014*; *Tzouvelekis et al., 2012*).

In Australia, carbapenemases have been rarely identified in Enterobacteriaceae, apart from the weakly carbapenem-hydrolysing $bla_{IMP-4}$ which has established low-level endemicity (*Australian Commission on Safety and Quality in Health Care, 2013*; *Australian Group on Antimicrobial Resistance, 2014*). Although small outbreaks and limited patient-to-patient transmission of CPE in Australia have been described (*Chang et al., 2015*; *Espedido et al., 2013*; *Kotsanas et al., 2013*; *Peleg et al., 2005*; *Tai et al., 2015*), the majority of patients identified with CPE are thought to have acquired these organisms during international travel to endemic countries (*Chua et al., 2014*; *Fernando, Collignon & Bell, 2010*; *Poirel et al., 2010*; *Sidjabat et al., 2011*, *2013*). Limited transmission in the community has also been reported (*Blyth, Pereira & Goire, 2014*).

*Klebsiella pneumoniae* carbapenemases (KPCs) were first identified in the United States of America in 1996, but have since spread worldwide to be endemic in several countries including the USA, Israel, Italy, Greece, Brazil and China (*Munoz-Price et al., 2013*). Several gene variants have been reported from the KPC family, though the most commonly reported alleles are KPC-2 and KPC-3, that differ by a single amino acid substitution (*Tzouvelekis et al., 2012*). The first reported KPC-producing organism

(KPC-2) in Australia was isolated in 2010 from a patient who had returned to Sydney, New South Wales, after being hospitalised in Greece (*Coatsworth et al., 2012*). In 2012, KPC-2 was first identified in the state of Victoria in a patient who had been repatriated to a metropolitan hospital also after a prolonged admission in a Greek hospital (*Chua et al., 2014*).

Whole-genome sequencing (WGS) has emerged as a powerful tool for bacterial strain typing and outbreak investigation (*Kwong et al., 2015*), and has been used in public health to assess transmission of *Listeria monocytogenes* and other foodborne pathogens at jurisdictional, national and international levels (*Kwong et al., 2016*). It has also been used at a single institution level to investigate small-to-medium sized outbreaks of KPC-producing Enterobacteriaceae (*Jiang et al., 2015*; *Kanamori et al., 2017*; *Marsh et al., 2015*; *Mathers et al., 2015*; *Ruppe et al., 2017*; *Snitkin et al., 2012*; *Weterings et al., 2015*). While the details of who infected whom can be traced among a small group of individuals in conjunction with a detailed epidemiological investigation, the transmission dynamics of larger established outbreaks across multiple institutions are more difficult to resolve (*Gilchrist et al., 2015*).

Due to the increasing incidence of KPC-producing Enterobacteriaceae during 2012–2014 from multiple sources across the state of Victoria (approximate population six million), we undertook an outbreak investigation employing WGS in parallel with traditional outbreak epidemiology. At the time this combined genomics and epidemiological investigation was initiated in June 2014, half of the 30 cases reported were from two metropolitan hospitals that each reported prior internal outbreaks of KPC-producing *K. pneumoniae*—one in 2012 and the other in early 2014. However, no clear epidemiological links had been identified between the two hospitals. Additionally, the remaining cases not identified through those two hospitals comprised patients in seven other healthcare facilities and in the community. Although all but one of the isolates were ST258 by multi-locus sequence typing (MLST), if and where transmission of KPC-producing *K. pneumoniae* was occurring was unknown. Here we report the utility of advanced genomic approaches including sampling for within-host clonal diversity, Bayesian transmission modelling, and plasmid genome reconstruction to assess the transmission dynamics of the outbreak to address these questions and inform infection control management in real-time during the course of the outbreak.

## MATERIALS AND METHODS

An overview of the workflow and methods used is shown in Fig. 1.

### Laboratory and genomic methods

#### *Isolate selection and antimicrobial susceptibility assays*

From 2012, following the first KPC isolate in the state, all Victorian diagnostic microbiology laboratories were asked to refer suspected carbapenemase-producing isolates to the Microbiological Diagnostic Unit Public Health Laboratory (MDU PHL) for further testing. Suspected isolates were defined as Enterobacteriaceae with a meropenem minimum inhibitory concentration (MIC) $\geq 0.5$ mg/L or zone diameter (ZD) $\leq 23$ mm,

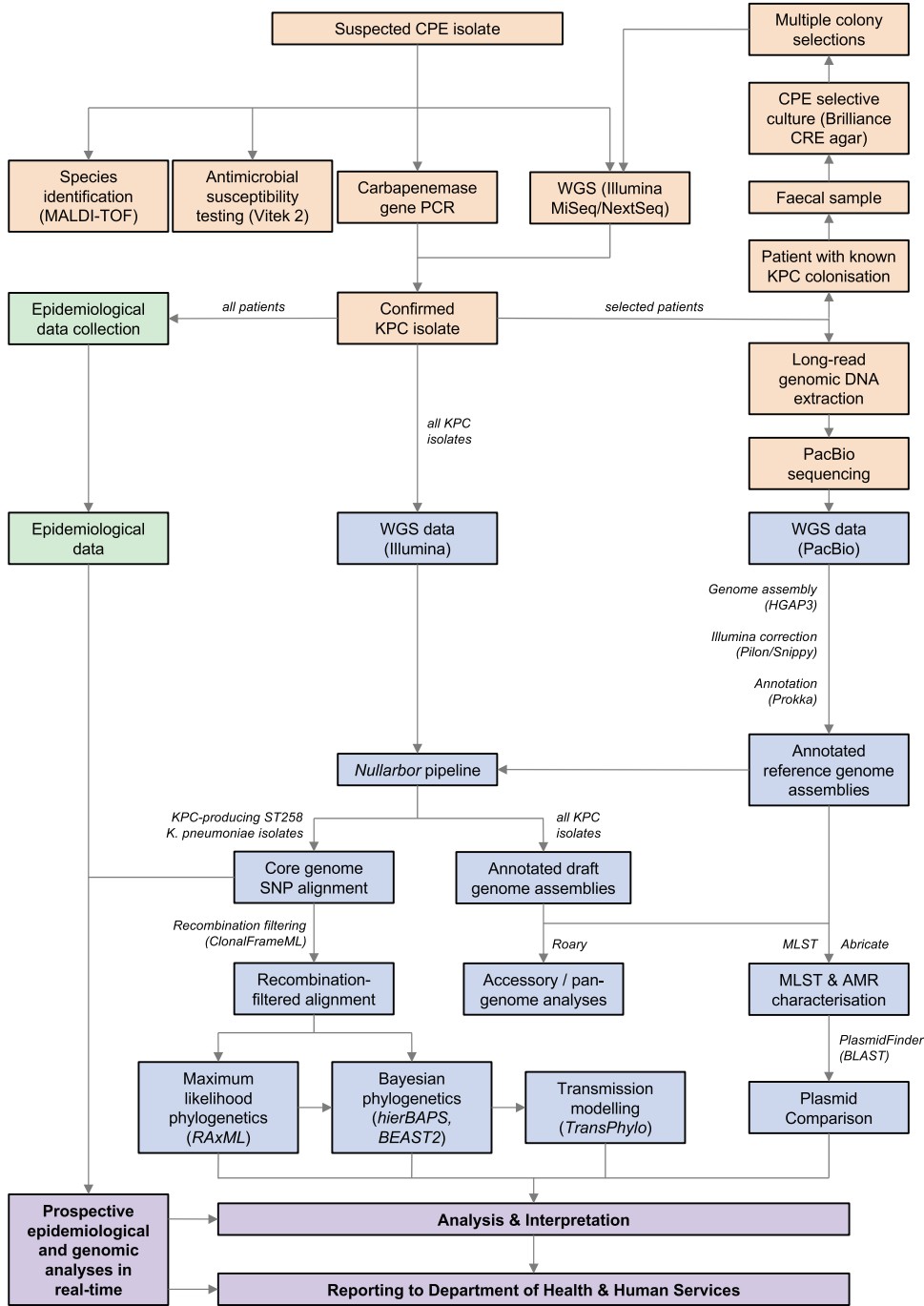

**Figure 1 Workflow summary diagram of methods used in this study.** Microbiology methods (orange), bioinformatic methods (blue), and epidemiological methods (green) used to generate data for analysis and reporting to public health authorities (purple) are shown. Results of combined prospective epidemiological and genomic analyses performed iteratively during the outbreak were reported to the Department of Health and Human Services in real-time. CPE, carbapenemase-producing Enterobacteriaceae; KPC, *Klebsiella pneumoniae* carbapenemase; MALDI-TOF, matrix-assisted laser desorption/ionisation time-of-flight; PCR, polymerase chain reaction; WGS, whole-genome sequencing; SNP, single nucleotide polymorphism; AMR, antimicrobial resistance; MLST, multi-locus sequence typing.

or an imipenem MIC $\geq 2$ mg/L or ZD $\leq 23$ mm. Isolates with a positive colorimetric (e.g. Rapidec® CarbaNP; bioMérieux, Marcy-l'Étoile, France (*Nordmann, Poirel & Dortet, 2012*)) or molecular (e.g. GeneXpert®; Cepheid, Sunnyvale, CA, USA (*Tenover et al., 2013*)) test were also requested. Duplicate KPC-producing Enterobacteriaceae isolates of the same species from a patient during a single hospital admission were usually excluded by referring laboratories. However, for some patients, multiple isolates were referred to MDU PHL if they originated from different laboratories, and were subsequently included in the analyses.

Species identification was confirmed using matrix-assisted laser desorption/ionisation time-of-flight (MALDI-TOF) mass spectrometry (VITEK® MS; bioMérieux, Marcy-l'Étoile, France) with routine susceptibilities tested on selected isolates using VITEK® 2 Compact (AST-N246 cards; bioMérieux, France). All suspected CPE isolates meeting the above criteria underwent testing for carbapenemase genes using polymerase chain reaction (Table S1). Isolates confirmed as KPC-producing organisms underwent WGS and were included in the genomic analyses, with the respective patients included in the epidemiological investigation. Isolates obtained prior to 2014 were sequenced retrospectively, while WGS was performed prospectively during the outbreak on isolates collected from 2014 onwards.

### Multiple colony sampling

To understand the genomic diversity present within a single individual and assess the influence on reconstructing transmission networks, primary faecal specimens were obtained from one patient. After overnight culture on Brilliance™ CRE selective media (Thermo Fisher, Waltham, MA, USA), 10–15 colonies, including any that differed in colony morphology, were selected from the plates for DNA extraction and WGS. Multiple colony sampling and sequencing for other patient samples was not performed due to cost limitations.

### DNA extraction and WGS

Bacterial cultures were purified for DNA extraction by two successive single colony selections after streaking onto horse blood agar incubated overnight at 37° C. DNA was extracted from a liquid suspension of the purified cultures using the QIAmp DNA Mini Kit (Qiagen, Hilden, Germany) or the JANUS Chemagic Workstation with the Chemagic Viral DNA/RNA kit (CMG-1033; PerkinElmer, Waltham, MA, USA).

Whole-genome sequencing was performed on the Illumina MiSeq or NextSeq platforms using Nextera XT libraries and protocols (Illumina, San Diego, CA, USA) with a minimum average quality score of 30 and a target sequencing depth of $\geq 50\times$ as previously described (*Carter et al., 2016*). Isolates not meeting these metrics were resequenced. Raw sequence data has been uploaded to the Sequence Read Archive under BioProject PRJNA397262.

Single molecule real-time (SMRT) sequencing of one *K. pneumoniae* and one *Citrobacter farmeri* isolate was also performed on the PacBio RS II platform (Pacific Biosciences, Menlo Park, CA, USA) using previously reported methods (*Carter et al., 2016*).

In brief, genomic DNA was extracted using a GenElute™ Bacterial Genomic DNA kit (Sigma-Aldrich, St. Louis, MO, USA) and SMRTbell libraries were prepared using manufacturer protocols (Pacific Biosciences, Menlo Park, CA, USA). Libraries underwent an additional size-selection step using a 20 kb template BluePippin size-selection protocol (Sage Science, Beverly, MA, USA).

### Bioinformatic analyses

Illumina raw sequencing reads were trimmed to clip Nextera adapters and low-quality sequence (Phred score <10) using *Trimmomatic* v0.36 (*Bolger, Lohse & Usadel, 2014*). The trimmed reads were assembled de novo with the *SPAdes* v3.7.1 assembler (*Bankevich et al., 2012*) and auto-annotated with *Prokka* v1.12-beta (*Seemann, 2014*). The SMRT analysis portal v2.3.0.140936 (Pacific Biosciences, Menlo Park, CA, USA) was used for isolates sequenced on the PacBio RS II, with raw sequence reads assembled de novo using the HGAP3 protocol, and error correction and polishing with *Quiver* v1. Further error correction was performed by mapping Illumina short reads from the same isolate to the PacBio assembly with *Pilon* v1.21 (*Walker et al., 2014*) and *Snippy* v3.2 (https://github.com/tseemann/snippy). The completed genome assemblies have been uploaded to GenBank under BioProject PRJNA397262.

From the annotated assemblies, the multi-locus sequence type (MLST) was determined in silico as was the presence or absence of antimicrobial resistance ('resistome') and other genes (pan-genome) using BLAST-based tools (https://github.com/tseemann/mlst) (https://github.com/tseemann/abricate) (https://github.com/sanger-pathogens/Roary) (*Page et al., 2015*). The pan-genome data was imported into *FriPan* (https://github.com/drpowell/FriPan) for web-browser visualisation using a Python script (https://github.com/kwongj/roary2fripan), and annotated genomes were visualised in *Geneious* v7.1.5 (http://www.geneious.com/).

Sequencing reads were also aligned to a reference genome to produce a reference-based whole-genome alignment including single nucleotide polymorphism (SNP), invariant and missing sites (https://github.com/tseemann/snippy). This alignment was then trimmed to exclude plasmid sites (https://github.com/kwongj/trim-aln). Putative regions of recombination were predicted using *ClonalFrameML* v1.0 (*Didelot & Wilson, 2015*), and masked in the alignment (https://github.com/kwongj/cfml-maskrc). Core genome SNP sites were extracted from the recombination-filtered alignment (*Page et al., 2016*) and a maximum likelihood phylogenetic tree inferred from the resulting SNP alignment in *RAxML* v8.2.4 (*Stamatakis, 2014*), using a General time reversible model of nucleotide substitution with a $\Gamma$ model of rate heterogeneity and four rate categories, with 1,000 bootstrap replicates to determine branch support. Hierarchical Bayesian analysis of population structure (*hierBAPS*) (*Cheng et al., 2013*) was used to provide further support for identifying phylogenetic clades, with clustering performed using eight levels in the hierarchy (L) and the prior maximum number of clusters (maxK) set at 10. Publicly available sequencing data were retrieved from the National Center for Biotechnology Information (NCBI) Sequence Read Archive and GenBank for comparison with local genomes (Tables S2 and S3). Three reference genomes were used. The initial genomic

analysis was performed using KPNIH24 (GenBank accession CP008797), with PacBio sequences of internal isolates (*K. pneumoniae* AUSMDU00008079, GenBank accession SAMN07452764; and *C. farmeri* AUSMDU00008141, GenBank accession SAMN07452765) used for subsequent analyses to minimise inaccuracies and biases from using distant genomes and draft genome assemblies as references (Fig. S1).

### Genomic context of the bla~KPC~ gene

The genomic context of the *bla*~KPC~ gene was investigated using a custom Python script (https://github.com/kwongj/contig-puller) to extract and align de novo assembled contigs carrying the gene. The flanking Tn*4401* transposon regions were compared to previously described isoforms (*Chen et al., 2012*; *Kitchel et al., 2009*; *Naas et al., 2008*, *2012*), using *BLAST+* (*Camacho et al., 2009*). Plasmid replicons were identified in genome assemblies using *BLAST+* against the *PlasmidFinder* database (*Carattoli et al., 2014*). Plasmids were presumed to carry KPC if *bla*~KPC~ was identified on the same contiguous sequence of DNA as the plasmid replicon.

### Bayesian evolutionary analyses and transmission modelling

Bayesian evolutionary analyses were also conducted to ascertain if the genomic signal could estimate the date of emergence of any outbreak clusters, and if any additional data could help inform where and when transmission events were taking place. A recombination-filtered chromosomal alignment was obtained as described above. The subsequent alignment was used as input into *BEAST2* v2.4.3 (*Bouckaert et al., 2014*) with collection dates entered for each isolate. The relationship between root-to-tip distance and date of isolation was assessed in *TempEst* v1.5 (*Rambaut et al., 2016*). We fitted a model with a relaxed log-normal clock (*Drummond et al., 2006*) to the alignment to account for inter-clade variation, assuming a HKY model for nucleotide substitution with $\Gamma$ distributed among site rate variation (*Hasegawa, Kishino & Yano, 1985*), and used a constant population size coalescent prior on the genealogy. Nodes were selected for logging the likely time of the most recent common ancestor (MRCA) for major clades identified in the maximum likelihood tree. We used eight Markov chain Monte Carlo (MCMC) runs of 100 million states, with sampling every 5,000 states, and a burn-in of 50%. The posterior samples from each chain were checked in *Tracer* v1.6 (http://tree.bio. ed.ac.uk/software/tracer/) for convergence, and then grouped into a single chain. The posterior samples for the dates of the nodes of interest were annotated on a maximum clade credibility tree and exported.

The maximum clade credibility tree was used as input into *TransPhylo* v1.0 (*Didelot et al., 2017*) to reconstruct the transmission chain. We based our prior for generation time (the time from primary infection of an individual to any subsequent secondary infection) on a previous study on the duration of carriage of KPC-producing Enterobacteriaceae (*Zimmerman et al., 2013*), assuming detection of KPC as an indicator of likelihood for onwards transmission. We therefore used a Gamma distribution with shape parameter 1.2 and scale 1.0 for the generation time prior, with a distribution mean of 1.2 years (438 days), standard deviation 1.096 years (400 days), and mode 0.2 years (73 days).

The MCMC was run for 100,000 iterations, with transmissions inferred from a consensus transmission tree.

### Outbreak genomic investigation

An initial retrospective analysis using the bioinformatic methods described above was performed in June 2014 on 41 KPC-producing *K. pneumoniae* isolates that had undergone WGS, including de novo genome assembly, in silico MLST, resistance gene detection and phylogenetic analysis. Subsequently, a customised, in-house pipeline was developed to streamline and automate handling of sequence data for ongoing assessment and analysis (https://github.com/tseemann/nullarbor). A summary report was generated for each set of isolates run through Nullarbor including quality control metrics, MLST, resistome and pan-genome comparison, and a maximum likelihood phylogeny inferred from the core SNP alignment (*Price, Dehal & Arkin, 2010*).

From this initial report, clonal complex (CC) 258 isolates underwent further analysis using a PacBio-assembled reference genome of a local isolate for a higher resolution core genome SNP alignment and phylogenetic comparison. As additional KPC-positive isolates were identified over time, the *Nullarbor* pipeline was repeated in an iterative process to establish the genomic relationship of new isolates to existing isolates, and where new CC258 isolates were identified, the CC258-specific analysis was also repeated with the new isolates. These analyses were interpreted together with epidemiological data in real-time as new cases emerged (see below—'Outbreak reporting and oversight'). Additional detailed analyses, including recombination filtering and Bayesian temporal analyses, were performed ad hoc to gain an overall understanding of the outbreak, and although were not used to directly inform infection control of individual cases, were used to inform overall outbreak management and guideline development.

## Epidemiological methods

Following the results of the initial genomic analysis and concern regarding local transmission, the ongoing genomic investigation was accompanied by collection of detailed epidemiological data, retrospectively for cases detected prior to 2014 and in real-time from 2014.

### Data collection

All patients from whom KPC-producing Enterobacteriaceae were identified from the first isolate in 2012 to 31 December 2015 were included in the epidemiological investigation. A case report form was developed to collect detailed epidemiological data, especially regarding patient hospitalisation details. Patients, or their next of kin, were interviewed by phone to ascertain demographic information on age, sex, country of birth, and risk factors for CPE such as hospitalisation and medical procedures, overseas travel, and comorbidities. Where a patient was in a healthcare facility at the time of specimen collection, or reported hospitalisation in the 12 months prior to initial CPE identification, medical records from each hospitalisation were examined to obtain data on specimen collection, clinical details of infection or colonisation, procedures and patient movements both before and after CPE identification, where available. Treating doctors, general

practitioners and infection control personnel were contacted to obtain additional and/or missing information. Patient hospital admission data were collated and used to identify putative transmission networks of patients linked by proximity in time and space.

Once identified with KPC, patients were assumed to be colonised indefinitely, thus subsequent KPC isolates were deemed to constitute the same patient episode and epidemiological data were not re-collected. KPC-producing Enterobacteriaceae isolated from normally sterile sample sites, e.g. blood, cerebrospinal fluid, pleural fluid, were considered to represent 'infections'. Isolates obtained from non-sterile sites (e.g. urine, wound swab, aspirate from intra-abdominal collection) where clinical evidence of infection was present and the patient was treated with antibiotics with activity against KPC-producing organisms, or where the treating clinician identified infection but elected to palliate the patient, were also considered to represent 'infections'. Other isolates were considered to represent 'colonisations', unless no epidemiological data were available ('unknown').

### Infection control investigation

For patients where local transmission of KPC-producing Enterobacteriaceae was suspected, interviews were conducted with infection control practitioners (ICPs) at the facility with putative transmission. Data were collected on environmental and contact screening activities, patient follow up and patient management alerts, isolation and collection of subsequent screening specimens.

### Outbreak reporting and oversight

Data from both genomic and epidemiological investigations were analysed together in real-time, and reported back to the Victorian Department of Health and Human Services. Where transmission events were recognised, the healthcare institutions involved were also informed. An incident management team was established through the Department of Health and Human Services to oversee the investigation and management of the outbreak.

### Ethics approval and consent to participate

Data were collected as part of an outbreak investigation through the Victorian Department of Health and Human Services under the Public Health and Wellbeing Act 2008 (https://www2.health.vic.gov.au/about/legislation/public-health-and-wellbeing-act). No consent was required for the project.

### Availability of data and materials

The raw Illumina short-read sequencing data and completed PacBio genome assemblies supporting the conclusions of this article are available in the NCBI database under BioProject PRJNA397262. In-house scripts used in the bioinformatics analyses are available at https://github.com/tseemann and https://github.com/kwongj.

## RESULTS

A total of 86 KPC-producing Enterobacteriaceae were referred to MDU PHL between 2012 and 2015 from 69 patients in 26 different healthcare facilities, six general practices
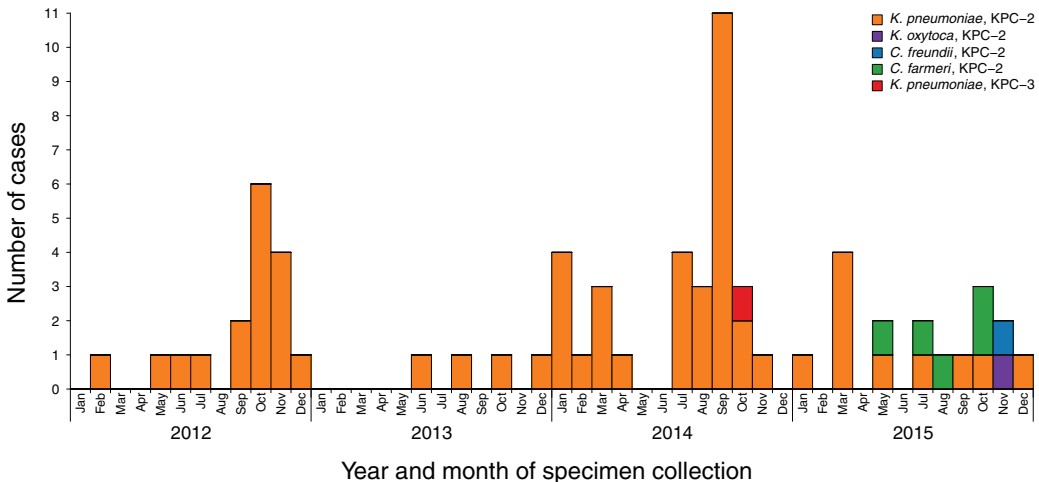

**Figure 2 Incidence of new KPC-producing Enterobacteriaceae cases referred to MDU PHL, 2012–2015.** Blocks are coloured by the species and KPC allele of the referred isolates. Repeated detections of KPC-producing isolates from the same patient have been excluded.

and three aged-care facilities across both metropolitan and rural Victoria (Fig. 2). Specimen source was reported for 85 of the isolates, with most specimens being urine (*n* = 42; 49%), faeces or rectal swabs (*n* = 20; 23%), or blood cultures (*n* = 10; 12%) (Table S4). Almost all isolates were KPC-2-producing Enterobacteriaceae, apart from two KPC-3-producing *K. pneumoniae* isolates from one patient. Overall, 79 of the isolates were *K. pneumoniae*, five were *C. farmeri*, with single isolates of *Klebsiella oxytoca* and *Citrobacter freundii*.

## Initial genomic analysis

In the initial retrospective genomic analysis of isolates collected prior to June 2014, 40/41 (98%) *K. pneumoniae* isolates collected from 29 patients were CC258 by in silico MLST, with all of these belonging to clade 1 of sequence type (ST) 258 *K. pneumoniae* described by *Deleo et al. (2014)*, based on analysis of the capsular polysaccharide gene island. Phylogenetically, these isolates clustered together in comparison to other previously reported international ST258 *K. pneumoniae* isolates (Figs. S2 and S3). Analysis of the local ST258 isolates revealed three distinct phylogenetic clades involving patients in 20 different healthcare locations, supported by pairwise SNP distributions and Bayesian analysis of the population structure (Fig. 3 and Fig. S4). These results formed the basis for ongoing prospective genomic analyses.

## Epidemiological analysis

A total of 57 patients were found to harbour isolates belonging to CC258 from 2012 to 2015. Most patients presented to the Emergency Department (*n* = 7) or were admitted to hospital (*n* = 44) at the time of initial specimen collection. Thirty-three (58%) were male, and the median age of the affected patients was 74 (range 20–94; IQR 62–83) (Table S5). Clinical infection was suspected in 61% of the cases, with the most common
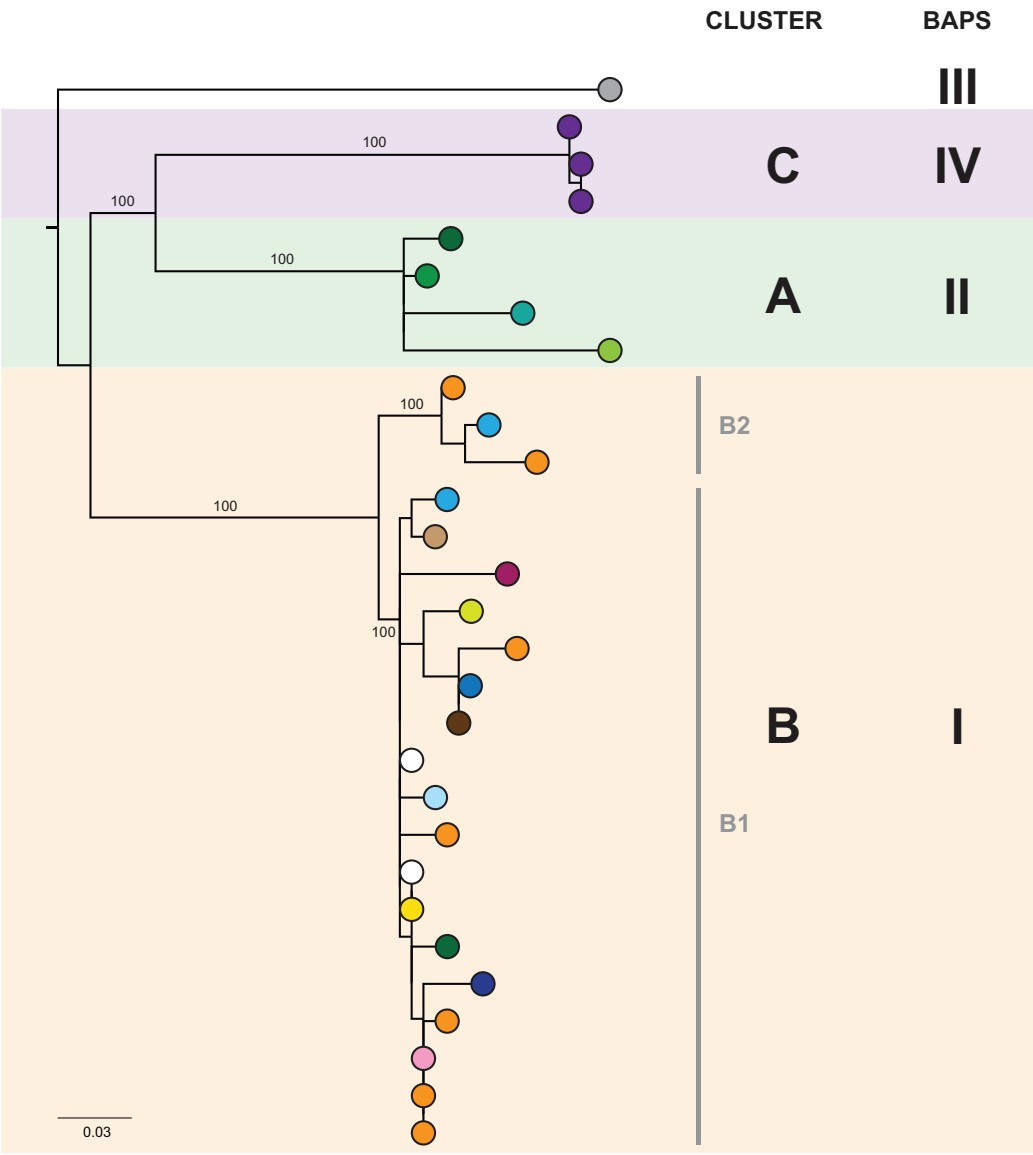

**Figure 3 The initial maximum likelihood phylogenetic tree comprised three dominant clades.** The tree includes 29 ST258 *K. pneumoniae* isolates collected from 29 patients prior to June 2014, with external nodes coloured according to healthcare facility at time of sample collection. Recurrent isolates from each patient have been excluded. Multiple patients in Facility A (purple) and Facility F (orange) were colonised/infected with KPC-producing *K. pneumoniae*, corresponding to known previous outbreaks in those hospitals. Support values (%) from 1,000 bootstrap replicates are shown for major branches. Major phylogenetic clades have been labelled cluster A (green shading), B (orange shading), and C (purple shading) in the order that the clades emerged, with the larger clade B comprising two subclades, B1 and B2. Corresponding clusters identified through Bayesian analysis of the population structure (BAPS) are also shown. The tree was rooted using an outgroup isolate (*K. pneumoniae* NJST258_1, GenBank accession CP006923.1; not shown in the tree) from a different ST258 clade.

presentation being urinary tract infection (*n* = 24; 42%). Sepsis was reported in 63% of those with infection due to KPC-producing *K. pneumoniae*. KPC-producing isolates were obtained from patients with clinical infection more commonly during 2012–2014 than 2015,

where >50% of the KPC-producing isolates identified were colonising or screening isolates (Fig. S5).

Of patients from whom a complete travel history was able to be obtained ($n = 46$), 22 (48%) reported overseas travel since 1996, and only eight (14%) reported travel in the 12 months prior to hospitalisation, strongly reinforcing the suspicion of local transmission of KPC. All 57 patients were found to have been hospitalised in Australian healthcare facilities in the 12 months prior to initial positive specimen collection, with a median length of stay of 61 inpatient days (range 0–181 days; IQR 34–101). Four patients had fewer than 10 inpatient days in an Australian healthcare facility during this period, three of whom had recent overseas hospitalisation, for which complete hospitalisation data could not be collected. Additional epidemiological data is detailed in the Supplemental Appendix.

A detailed analysis of patient admission data was undertaken in attempt to identify location or source of acquisition. Forty-one patients (72%) had previously attended one facility (Facility F) in the 12 months prior to initial identification of a KPC-producing isolate. However, the remaining 16 patients reported no hospitalisation in this facility, and 18 additional facilities were identified in which two or more patients had been admitted. Three putative transmission networks were identified based on overlapping admissions to the same hospital ward at the same time among patients detected with KPC. It became apparent that due to the sheer number of hospitalisations and the complexity of patient movements, drawing firm conclusions on who transmitted to whom and where transmission had occurred solely from epidemiological data would be difficult (Fig. 4 and Fig. S6).

## Combined analysis of CC258 KPC-2-producing *K. pneumoniae* isolates

Despite identification of three distinct phylogenetic clades in the initial genomic analysis (Fig. 3), we were unable to exactly define where transmission was occurring as the samples were distributed across multiple locations. Of the three clades (designated as genomic cluster A, B and C), cluster C had the best correlation with an epidemiologically defined transmission network, comprising isolates from patients in the same healthcare facility at the same time (Fig. 4). However, other correlations were not immediately apparent. Cluster A included isolates from three patients in three different facilities that were within the same healthcare network and/or were geographically located in the same region within a 20 km radius, though were 30 km away from the fourth patient in a different healthcare network in the cluster. Cluster B included isolates from patients in 12 different locations across the state of Victoria, separated by up to 500 km. Seven isolates were obtained from patients attending local general practice (primary care) clinics.

Using the genomic data to enhance and allow flexibility in the epidemiological data (e.g. including admissions to adjacent wards or admissions separated by a few days as 'overlapping'), it became apparent that each of the genomic clusters corresponded to a separate transmission network (Fig. 5). The largest of these (cluster B) was located at a single institution (Facility F), and as further isolates were identified, two subclusters

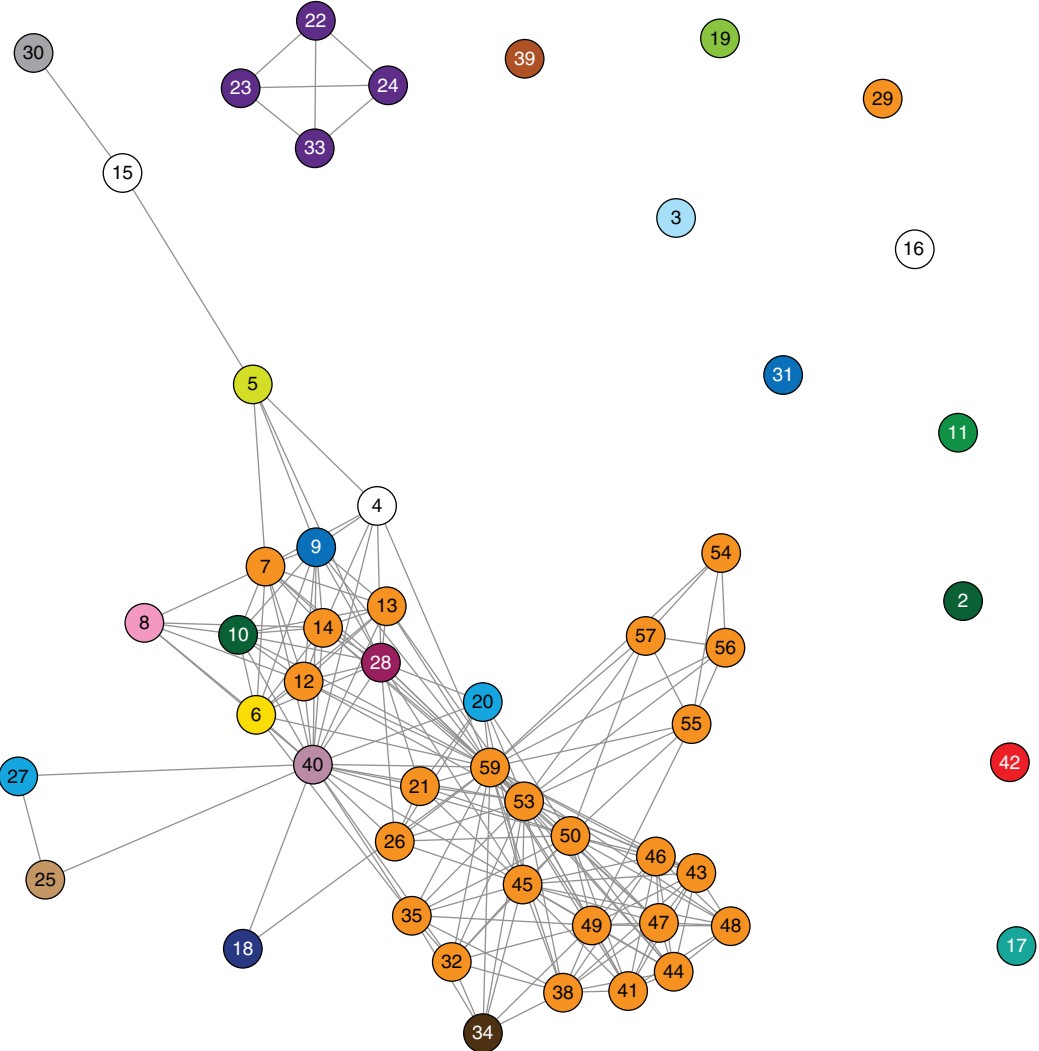

**Figure 4 Epidemiological links based on overlapping patient admissions were unable to resolve where and when transmission was occurring for most isolates.** Links in the above network connect patients based on overlapping patient admissions in the same hospital ward at the same time (minimum of one day overlap). Links that occurred after detection of $bla_{KPC}$ in both patients have been excluded. Nodes are numbered by patient, and coloured by healthcare facility (as in Fig. 3) at the time of sample collection. The epidemiological network connecting patients 22, 23, 24 and 33 correlates with the closely related genomes in cluster C (Fig. 3).

(B1, B2) were defined. Cluster B2 included isolates arising from suspected transmission in mid-2014 in an inpatient aged care ward, while cluster included isolates from a number of patients who were admitted to Facility F in 2012, though to several different wards. A second transmission network in cluster B1 was identified in another aged care ward in 2015. The predicted transmission network corresponding with cluster C was also confirmed, with the isolate genomes differing by <5 SNPs, and the corresponding epidemiological data for the respective patients showing overlapping hospital admissions to a single hospital ward in late 2013 (Fig. 6).
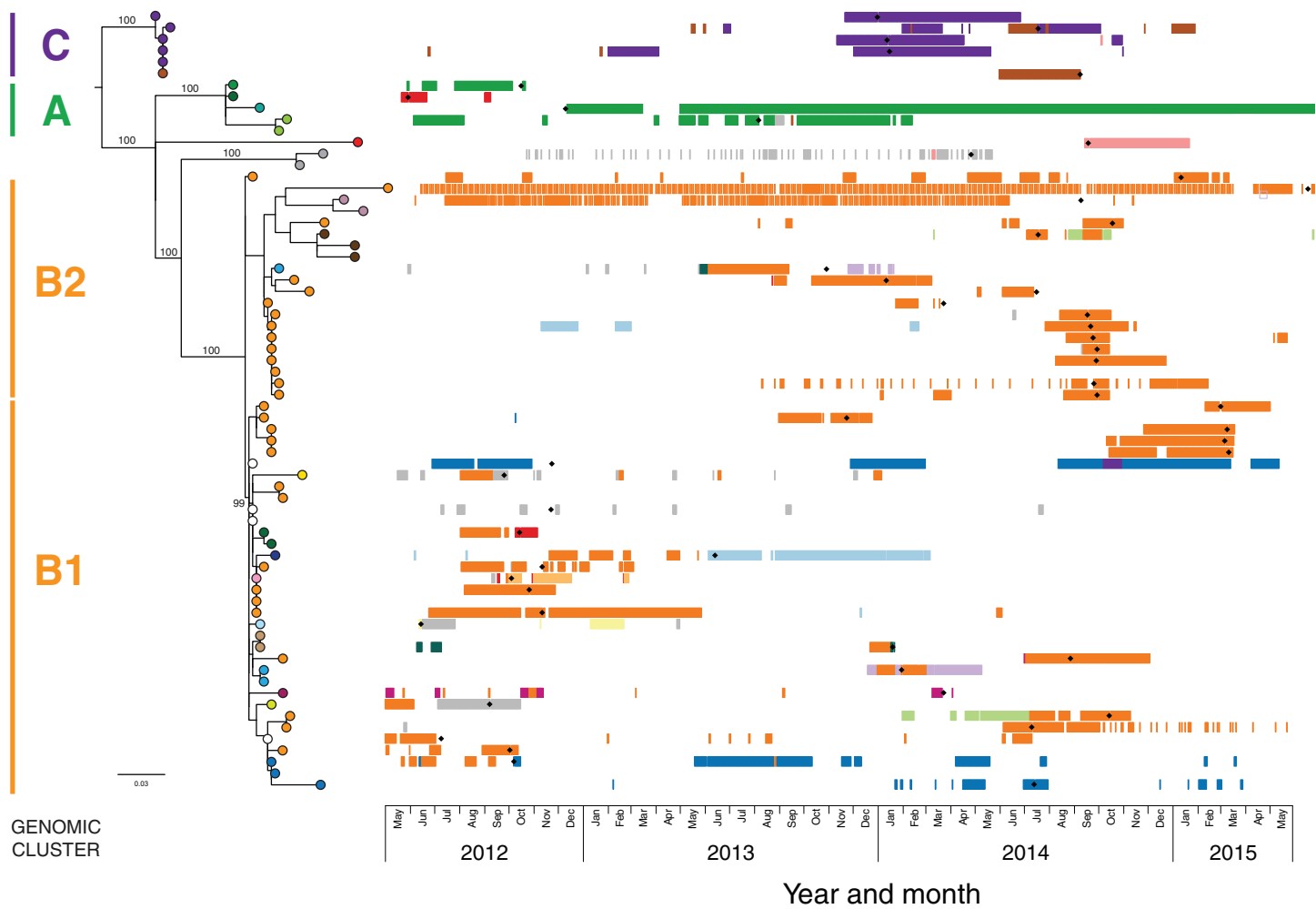

**Figure 5 Combined analysis, with genomic relationships between isolates overlaid upon epidemiological data, delineated multiple transmission networks.** A maximum likelihood phylogenetic tree is shown on the left, labelled with major genomic clusters and supporting branch bootstrap values (%) from 1,000 replicates for major branches. Nodes of the tree are coloured by healthcare facility at the time of sample collection as in Figs. 3 and 4. Coloured horizontal bars on the right indicate healthcare facility admissions over time (x axis), with different colours representing different healthcare networks. Black diamonds (♦) indicate first detection of KPC for each patient.

Combining epidemiological and genomic data also revealed secondary transmission events, where patients with undetected colonisation at one hospital were transferred to a second healthcare service, with subsequent diagnosis and onwards transmission. Patients whose isolates were phylogenetically clustered with other isolates from a known transmission network, but who had not previously been admitted to the hospital where transmission was occurring, were flagged as potentially having acquired KPC through a secondary transmission event. Through this, two putative secondary transmission events were later identified from these main transmission networks (see example case study in Fig. 6).

### Bayesian evolutionary and transmission analysis

Although from the combined genomic and epidemiological analyses, it appeared that local transmission within separate hospitals was driving the outbreak, local isolates from

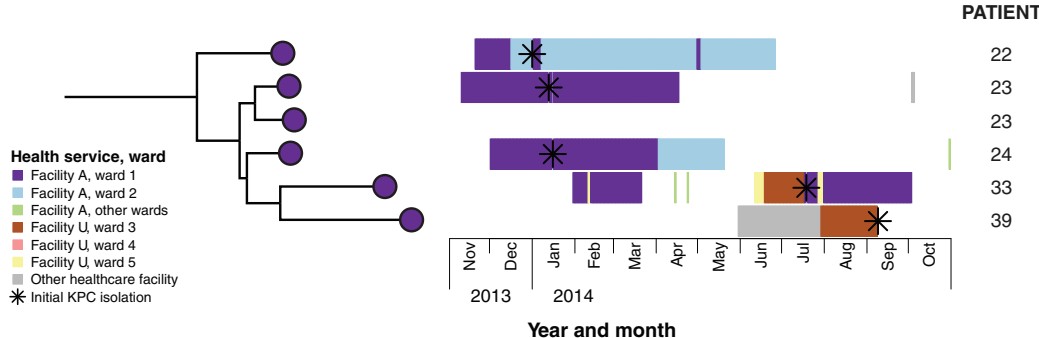

**Figure 6 Combined analysis of genomic and epidemiological data of the cluster C network identified secondary transmission.** Example case study: Patients 22 and 23 both reported overseas hospitalisation in the 12 months prior to first detection of KPC—patient 22 in Vietnam for spinal surgery following a motor vehicle accident, and patient 23 in Greece for stem cell therapy. Both patients had undergone rectal screening on admission to Ward 1 in Facility A, with patient 22 being placed in intensive contact precautions for the duration of his hospital admissions after isolating another multidrug-resistant organism, though was required to use shared bathroom facilities with patients in the adjacent room. Having required treatment with meropenem for both hospital-acquired pneumonia and a surgical wound infection, patient 22 was later diagnosed with a KPC-producing *K. pneumoniae* indwelling catheter-associated urinary infection in January 2014. Twelve days later, patient 23 was subsequently found to have a polymicrobial sacral wound infection, with cultures including KPC-producing *K. pneumoniae* from sacral tissue. In response to this, all patients on the ward who had been admitted to the same room and/or shared bathroom facilities with patients 22 and 23 were screened, with the subsequent identification of patient 24. Alerts were placed on the records of patients meeting the criteria who had been previously discharged. Environmental screening of the rooms and bathrooms was conducted, with no KPC-producing organisms identified, and extended bleach cleaning with changes of curtains, chairs and other furnishings was conducted for the entire ward. Patient 33 was also admitted to Ward 1 in Facility A in February 2014, subsequent to the identification of KPC-2 in patients 22, 23 and 24. This patient was not screened as he had not been admitted to the same room, nor had he shared a bathroom with the identified cases. He also reported no recent history of overseas travel. However, he was identified in July 2014 through routine screening at Facility A following transfer from Ward 3, Facility U, located 25 km away. A KPC-producing isolate from patient 39 was identified in September 2014, and although the isolate genomically clustered with isolates from patients 22, 23, 24 and 33 identified at Facility A, she had no previous presentations to that healthcare facility. However, immediately prior to identification of KPC, she had also been in Facility U on Ward 3, though she was admitted there 13 days after patient 33's discharge. Given this was the only plausible epidemiological link to the other cluster C patients, secondary transmission was presumed to have occurred in Ward 3, Facility U.

each cluster were closer to other local isolates (median pairwise SNP distance = 16; IQR 9–38) than to international ST258 isolates from GenBank (median pairwise SNP distance = 243; IQR 80–258). To investigate the possibility of initial inter-hospital transmission leading to subsequent transmission networks within each hospital, we explored whether Bayesian evolutionary and transmission modelling could be used to provide additional insight.

From these analyses, genomic clusters A and C each appeared to be derived from separate overseas importations of KPC-producing *K. pneumoniae* with subsequent spread within a single hospital or healthcare network rather than inter-hospital transmission, with a MRCA predicted to have occurred prior to 2010 (median 2007.78; 95% HPD 2003.30–2011.09) when the first KPC-producing organism was reported in Australia (*Coatsworth et al., 2012*), and 2012, when the first KPC-producing organism was reported

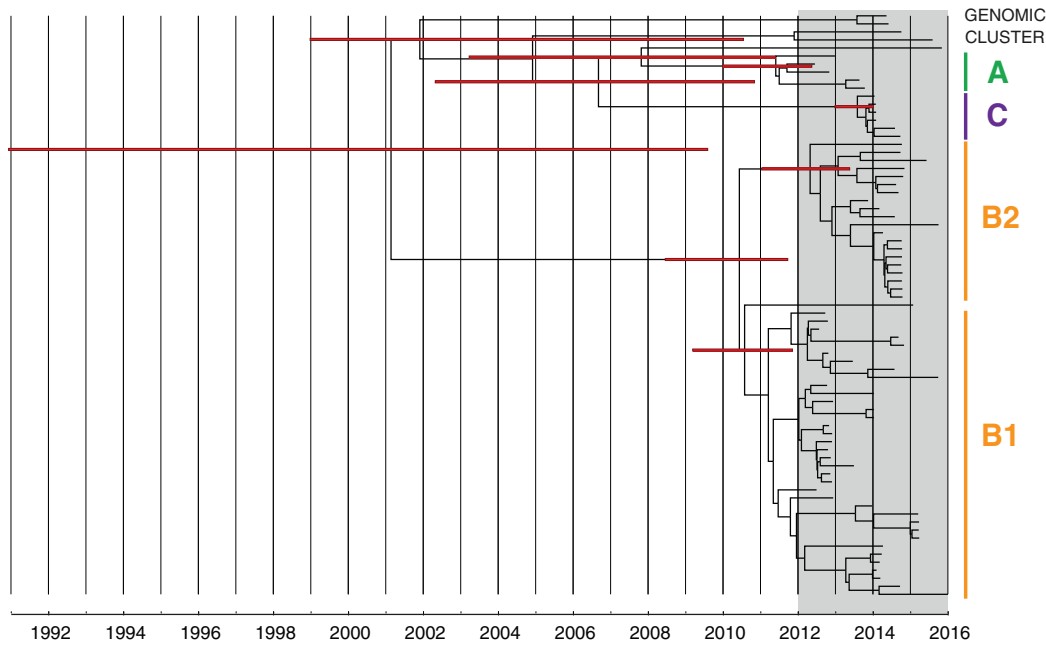

**Figure 7 Bayesian evolutionary analysis indicates each of the phylogenetic clades corresponding to the genomic clusters emerged prior to the detection of KPC in Victoria.** A maximum clade credibility timed phylogeny from Bayesian evolutionary analysis of local CC258 *K. pneumoniae* isolate genomes are shown, with median node heights displayed. The thin red bars indicate 95% highest posterior density (HPD) intervals for the most recent common ancestor (MRCA) for major clades and defined genomic clusters (indicated on the right). The shaded grey region indicates the recent period when KPC isolates were detected in Victoria.

in Victoria (*Chua et al., 2014*) (Fig. 7). Similarly, the unclustered isolates represented separate individual importations in these analyses, supporting the known history of recent overseas travel for these patients (Table S5).

Cluster B likely arose from another overseas importation (MRCA cluster A and B: median 2002.52; 95% HPD 1991.72–2008.79), though the timed phylogeny (Fig. 7) suggested strains of KPC-producing *K. pneumoniae* from cluster B may have been circulating in Victoria since 2011 (MRCA cluster B1 and B2: median 2010.59; 95% HPD 2008.67–2011.86). A number of internal nodes within cluster B1 were dated to 2012 (Fig. S8), corresponding with overlapping patient admissions at Facility F at that time, which affirmed the putative transmission network in Facility F in 2012. In both clusters B1 and B2, many isolates were linked by transmission events that were predicted to have occurred during 2013, despite a decline in new cases detected during this period. Notably, many of the isolates derived from these transmission events were not detected until later, in 2014 or 2015. This supported the epidemiological hypothesis that detection of previously unrecognised colonisation was driving a large proportion of the new cases in 2014 and 2015, rather than new transmission events (Fig. 8). For each of the suspected transmission networks identified in the combined outbreak investigation, posterior probability distributions for the MRCA were generally consistent with the epidemiological data (Fig. S8).

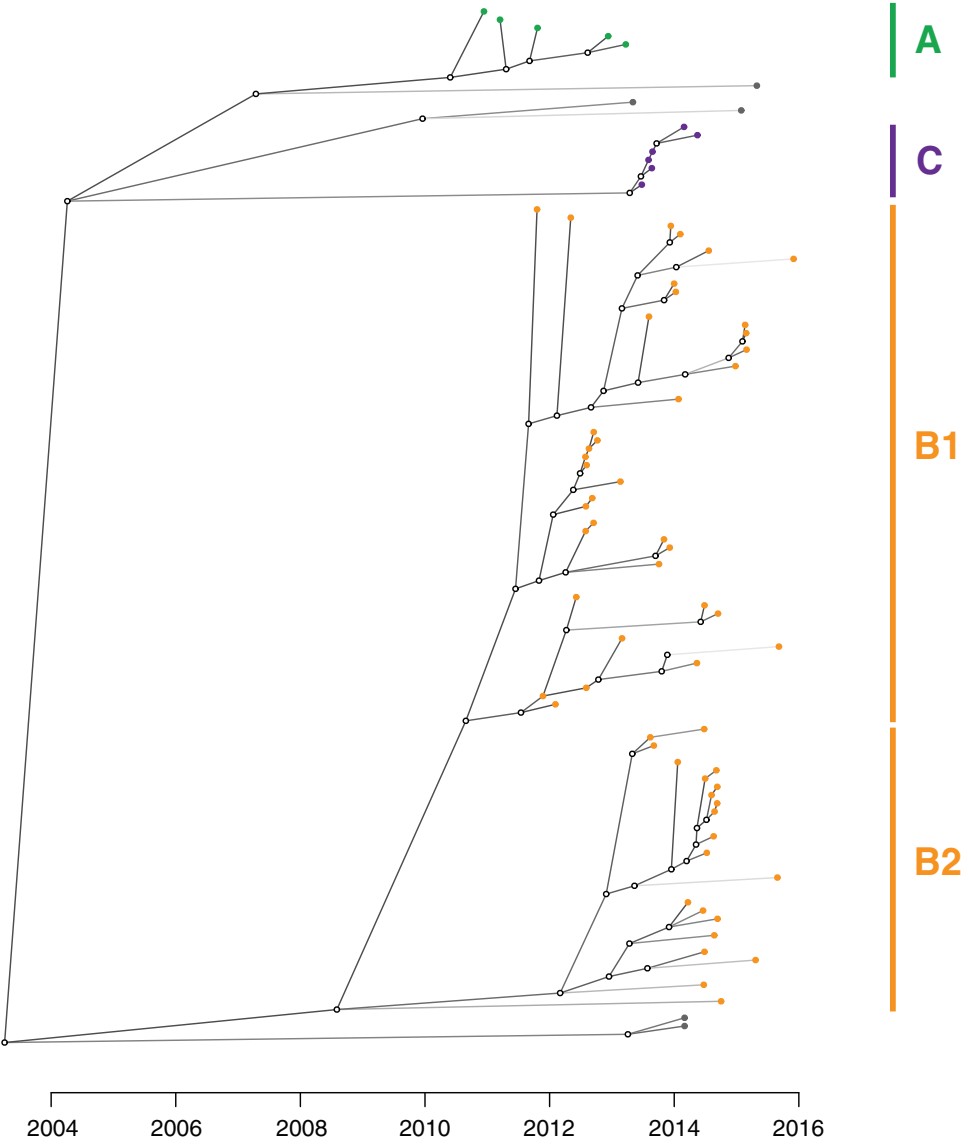

**Figure 8 An inferred transmission tree shows that undetected colonisation was significant in propagating the outbreak.** Solid nodes represent the posterior mean time of KPC acquisition by individuals and are coloured by the corresponding genomic cluster, with empty circles representing inferred unsampled individuals contributing to the transmission tree. Branches are shaded by number of missing links in the transmission tree, with lighter branches representing increasing numbers of missing links implicated.

From the *TransPhylo* analysis, the mean reproductive number ($R_0$) across the outbreak was calculated to be 1.63 (SD = 0.13), accounting for an average estimated sampling proportion of 0.27 (SD = 0.05).

## Plasmid analysis

In late 2015, other Enterobacteriaceae carrying $bla_{KPC-2}$ were detected, including *C. farmeri*, *C. freundii*, and *K. oxytoca* (Fig. 2), which raised concern about KPC-plasmid spread. Analysis of the de novo assembled draft genomes indicated the $bla_{KPC-2}$ carrying

plasmids in these organisms matched a $bla_{KPC-2}$ carrying plasmid found in the locally circulating *K. pneumoniae* isolates, suggesting inter-species plasmid movement, and subsequent organism transmission. All *K. pneumoniae* ST258 isolates were found to harbour $bla_{KPC-2}$ within the 'a' isoform of the characteristic Tn*4401* transposon, and usually on a contig containing sequence encoding the replication regulator RNAI from IncFII(K) plasmids. Analysis of the PacBio reference *K. pneumoniae* genome assembly showed the presence of both IncFIB(pQil) and IncFIB(K) plasmids. The ST258 draft genome assemblies also all contained IncFIB replicons, though the assemblies were inadequate to confirm the plasmid types carrying Tn*4401*. Chromosomal integration of the Tn*4401* within the putative diguanylate cyclase gene, *ycdT*, was found in two epidemiologically linked ST258 *K. pneumoniae* isolates. One ST258 *K. pneumoniae* isolate lacked $bla_{KPC-2}$, and subsequent repeat susceptibility testing confirmed the isolate to be susceptible to carbapenems. Re-testing of 10 colonies subcultured from the stored glycerol stock for that patient sample found only 7/10 displayed carbapenemase hydrolytic activity with the CarbaNP test, indicating in vivo or in vitro loss of the $bla_{KPC}$ gene.

Despite presence of the Tn*4401* and IncFII(K) RNAI in the *C. farmeri* draft assemblies, they lacked the IncFIB replicons detected in the *K. pneumoniae* genomes. To determine if interspecies plasmid transmission had occurred, one *C. farmeri* isolate underwent PacBio sequencing. Sequence comparisons between the KPC-plasmid genome from this isolate and the KPC-plasmid genome from the *K. pneumoniae* PacBio reference genome indicated almost identical plasmids. The *C. farmeri* plasmid carried an IncR-type replicon and lacked a Type I restriction modification system, but otherwise comprised an identical sequence to 85% of the *K. pneumoniae* KPC plasmid following a recombination and inversion event (Fig. 9). The *C. freundii* isolate was also found to have an IncR-type plasmid, but lacked other gene content found in the *C. farmeri* IncR plasmids, suggesting mobilisation of the Tn*4401* transposon. The exact location of $bla_{KPC-2}$ and plasmid type of the *C. freundii* and *K. oxytoca* isolates were unable to be determined due to the limitations of the assembled short-read sequencing data. Phylogenetic analysis of the *C. farmeri* isolates inferred from core genome (chromosomal) SNPs confirmed the isolates were closely related (Fig. S9).

### Within-host diversity

To assess the within-host genomic diversity of KPC-producing Enterobacteriaceae, the isolate genomes from 17 patients who had multiple isolates obtained and sequenced were compared. Of these, one patient had 32 isolates collected from seven samples over an eight-month period from February 2016 to October 2016, including multiple colony sampling from two faecal samples. The other 16 patients had multiple (median = 2; range 2–4) isolates referred to MDU PHL, obtained through recurrent presentations at different healthcare facilities.

Genomic comparisons indicated isolates from the same patient differed by up to 21 SNPs in the core genome. This included the 32 isolates obtained from patient 70, with several internal lineages emerging from the same common ancestor within that individual (Fig. 10). Pangenome analyses of these isolates also demonstrated changes

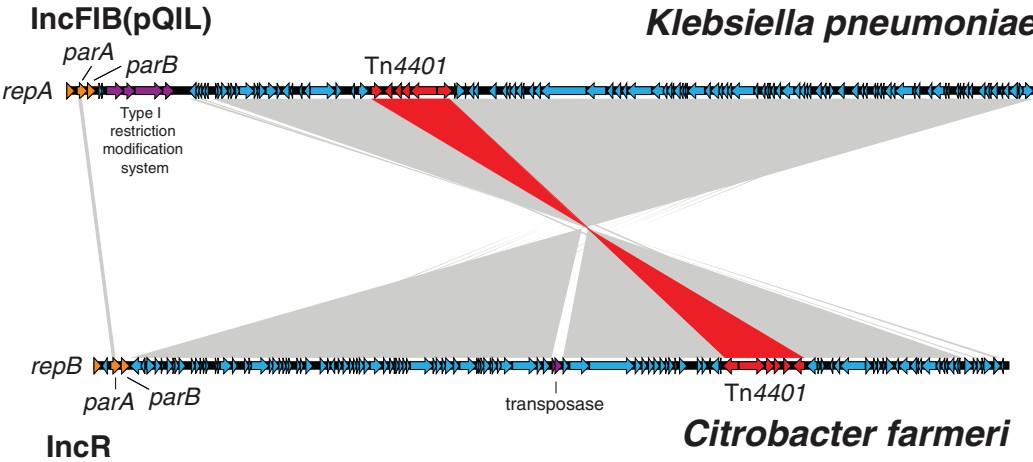

**Figure 9** *Klebsiella pneumoniae* carbapenemase plasmids from the *C. farmeri* isolates were almost identical to KPC plasmids from *K. pneumoniae* outbreak isolates, despite differing replication proteins. BLAST comparison between an IncFIB (pQIL-like) plasmid genome from a ST258 *Klebsiella pneumoniae* isolate, AUSMDU00008079 (above), and an IncR plasmid genome from a *Citrobacter farmeri* isolate, AUSMDU00008141 (below), from the outbreak. The grey shading indicates corresponding DNA regions of high nucleotide identity transcribed in opposing directions, with the Tn*4401* transposon harbouring $bla_{KPC-2}$ highlighted in red. The plasmid genomes have been orientated to their respective replicons and downstream plasmid partitioning genes, *parA* and *parB*.

in the accessory genome correlating with the emerging internal lineages in the core genome (chromosomal) phylogeny, and lineage-specific plasmid carriage (Fig. S10). These changes included subtle differences in plasmid types, acquisition/loss/switching of transposons, and loss/gain of heavy metal operons. There was a poor molecular clock signal for these isolates (root-to-tip divergence $R^2 = 0.026$), and no further temporal analyses were undertaken.

When compared to other outbreak isolates, multiple isolates from the same host clustered together within a single subclade in the phylogenetic tree. This supported the concept of a 'transmission bottleneck', where despite an assumed diverse within-host population of KPC-producing Enterobacteriaceae, transmission only involved a single isolate, with subsequent within-host evolution following the initial transmission event. This also suggested multiple acquisitions of different strains of KPC-producing *K. pneumoniae* by the same patient were uncommon among the patient cohort. Of patients with multiple isolates spanning a six-month period, only patients 18 and 70 appeared to have isolates from differing subclades, though bootstrap branch support for these subclades was marginal (0.70–0.75).

## Environmental diversity

A limited number of KPC-producing environmental *K. pneumoniae* isolates obtained at a single time-point from frequently touched surfaces in the room of patient 75 underwent WGS and were compared to the other patient isolates (Fig. 10). As expected, the environmental isolates clustered with the clinical isolate from patient 75, as well as those from patient 72, who had previously shared a (different) room with patient 75.

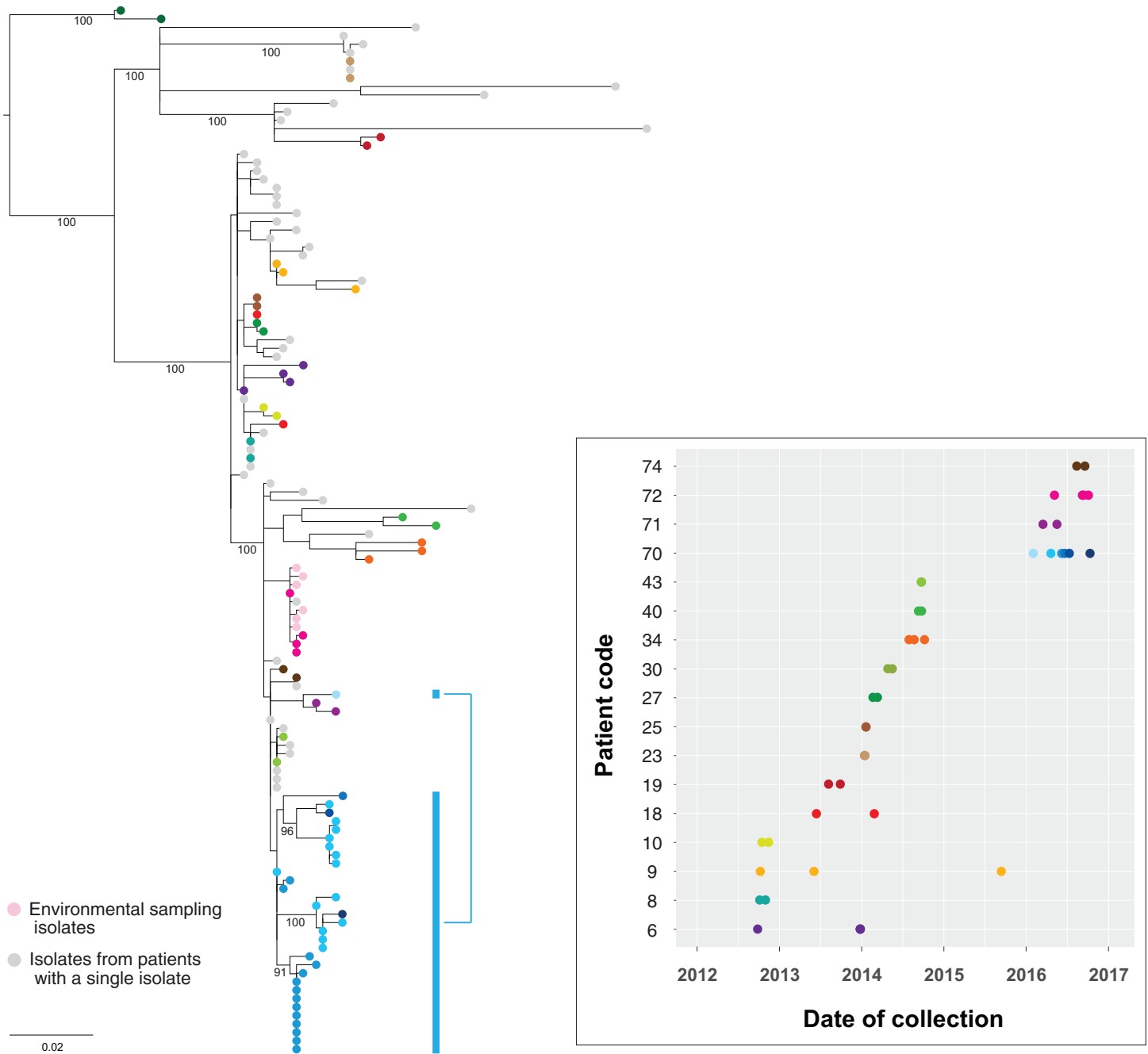

**Figure 10 Genomes of isolates from the same host group together in the phylogeny.** Maximum likelihood tree of the study isolates with additional isolates obtained up to 1 November 2016 included. Multiple isolates from the same patient have been coloured by patient, with the accompanying graph indicating the collection dates for the corresponding isolates. Thirty-two isolates from six clinical samples obtained from patient 70 (blue) over eight months have been highlighted to illustrate the within-host lineages emerging in this patient. Six environmental isolates from the room of patient 75 are also shown (light pink). Bootstrap values (%) from 1,000 replicates have been displayed for major branches in the tree.                                             

Notably, the genomic diversity between these clinical and environmental isolates was less than the within-host diversity seen within patient 70, supporting the hypotheses that the direction of transmission was from patient 75 to the environment, rather than

environment to patient, and that transmission had occurred between patients 72 and 75 when they previously shared a room.

## Infection control investigation

Qualitative interviews were conducted with ICPs at three facilities regarding five transmission networks (four primary clusters of KPC-producing ST258 *K. pneumoniae* including two secondary transmission events at different locations, in addition to one cluster of KPC-producing *C. farmeri*) of suspected local transmission involving 17 patients. The majority of transmission events were thought to have occurred due to unrecognised colonisation, and prior to the implementation of contact precautions— single room isolation with en-suite bathroom, with requirements for use of personal protective equipment (gloves and gown) by staff and visitors entering the room. However, two instances of apparent transmission to and from patients under contact precautions were reported. In each of these situations, after being admitted directly into contact precautions due to known colonisation with other multidrug-resistant organisms, a patient acquired a KPC-producing *K. pneumoniae* isolate that was highly related to other isolates in a genomic cluster (<5 SNPs in the core genome).

Due to the lack of local guidelines at the time of the outbreak, the management of wards where transmission was found to have occurred was at the discretion of the individual healthcare facilities involved. A retrospective review of the methods used to interrupt transmission identified a number of bundled approaches, with the common measures being contact screening, enhanced cleaning (in contrast to standard cleaning), and isolation of identified cases under contact precautions (Table S6).

## Impact of WGS on the outbreak investigation and development of local containment guidelines

In response to the genomic and epidemiological evidence of an evolving outbreak with local transmission, a state-wide management guideline was developed for the surveillance and containment of CPE incorporating the routine use of WGS to determine relatedness between isolates (*Department of Health & Human Services, Victoria State Government, 2015*).

Several insights were gleaned through the use of genomics that were fundamental to the outbreak investigation and informed development of the guidelines and their focus on hospitals. For example, through the epidemiological investigation, the identification of KPC in previously hospitalised patients suggested nosocomial rather than community transmission, though which hospital transmission was occurring within was not immediately clear, given the number of hospitals and prior overlapping hospitalisations involved. The resolution of the genomic data refined epidemiological hypotheses to pinpoint transmission areas in individual hospitals to focus infection control efforts, with Bayesian analyses of the genomic data also supporting the epidemiological findings. By determining when and where transmission was occurring, it was evident that many new cases were readmissions of patients with previously unrecognised colonisation, rather than new transmission events, making it difficult to 'control' the outbreak.

This highlighted the need to intensively screen at-risk patient contacts, including flagging those that had been discharged from an affected ward prior to recognition of transmission for screening and pre-emptive isolation upon subsequent hospital presentations. This real-time change-in-practice was reflected by the increasing proportion of cases identified as 'colonisations' rather than 'infections' due to intensified screening practices over time (Fig. S5).

## DISCUSSION

The emergence of CPE is a major threat to human health (*Australian Commission on Safety and Quality in Health Care, 2013*; *Centers for Disease Control and Prevention, US Department of Health and Human Services, 2013*; *World Health Organization, 2014*), with significant interventions required at state and national levels to contain the spread once established (*Schwaber & Carmeli, 2014*). Here, we describe the largest outbreak of CPE reported in Australia, and demonstrate how a combined genomics and epidemiological investigation delineated the outbreak into five separate nosocomial transmission networks (four clusters of *K. pneumoniae* and one cluster of *C. farmeri*) across the state over four years, resulting in targeted interventions for each transmission area. The use of WGS for outbreak analysis has been well established, including for transmission of multidrug-resistant hospital pathogens (*Palmore & Henderson, 2013*; *Snitkin et al., 2012*), but these studies have been predominantly small scale, single institution studies, and often retrospective. Genomics has rarely been used prospectively and in real-time during a complex multi-institutional outbreak requiring a coordinated state-wide public health response. Due to prolonged colonisation, many CPE cases linked to hospitals with suspected transmission were identified through other healthcare facilities, and some through general practice. With the number of potential transmission opportunities in retrospective hospitalisation data from all facilities, the epidemiological investigation would have been difficult to interpret and translate into focussed interventions without the resolution offered by genomics. Similarly, accurate interpretation of the genomic data would have been difficult without the supporting epidemiologic data. As others have pointed out, the two must go hand-in-hand (*Tong, 2013*). By integrating our detailed epidemiological investigation with genomic analyses, we were able to refine our hypotheses, and coordinate an effective public health response to target areas with ongoing transmission, emphasising the ability of WGS to enhance surveillance systems and outbreak investigations.

Despite integrating genomic data with detailed epidemiological information, we still encountered several challenges. Accurate inference of plasmid transmission can be challenging from short-read Illumina sequencing data, with repetitive elements such as insertion sequences that accompany mobile resistance elements frequently confounding short-read assemblers and read-aligners (*Conlan et al., 2014*). As we demonstrated here and as reported by others (*Conlan et al., 2014*), long-read sequencing technology such as Pacific Biosciences single-molecule sequencing can be highly useful in resolving and tracking the diversity of plasmids carrying carbapenemases and other resistance genes, although we could only sequence a limited representation of the outbreak isolates due to

cost limitations. The data gained from these completed genomes were invaluable. The use of a local internal reference provided additional confidence in SNP calling among highly clonal isolates. We were able to compare completely assembled plasmids between different species, which otherwise would not have been possible. Many isolates in the outbreak were hypothesised to carry $bla_{\text{KPC-2}}$ on either an IncFIB(K) or IncFIB(pQil) plasmid, with long-read sequencing showing it was possible for isolates to harbour multiple plasmids of the same incompatibility group. As we found in two isolates and as others have recently reported (*Mathers et al., 2017*), chromosomal integration of Tn*4401* also occasionally occurs. The movement of these resistance elements adds complexity to understanding transmission dynamics using genomics, and highlights the limitations of short-read sequencing data to fully characterise these elements.

It is increasingly evident that the subtly diverse populations of a single clonal type due to within-host evolution (*Golubchik et al., 2013*), and more diverse populations from transmission of mixed infections (*Eyre et al., 2013*; *Hatherell et al., 2016*), can impair accurate reconstruction of transmission pathways from genomic data (*Worby, Lipsitch & Hanage, 2014*). In our study, we found some individuals not only had different plasmid variants, but also had distinct within-host evolutionary lineages, indicating a complex evolutionary history of transmission, within-host evolution, and plasmid movement, mirroring recent reports in other patients with prolonged KPC colonisation (*Conlan et al., 2016*). However, measuring this diversity is difficult in outbreak investigations. Although single-colony sampling and sequencing arguably provides the most informative data, it may be subject to colony selection bias, so some uncertainty remains. This can be offset by sampling more colonies, but the time and cost of sequencing each colony can quickly become prohibitive in an outbreak investigation and real-world public health environment. Although some have attempted to detect the presence of mixed infections in short-read datasets through analysis of short-read mapping from deep sequencing (*Eyre et al., 2013*), this relies on WGS performed on a sweep of a primary culture plate, introducing the potential for exogenous DNA in the dataset, and becomes considerably more complicated with fluxes in accessory genome content. As we found, $bla_{\text{KPC}}$-plasmids can be lost even through laboratory passage of stored isolates. Even if the within-host genomic diversity can be captured, incorporating these data into models can be difficult. Recent attempts to account for elements such as within-host diversity and unsampled data in reconstructing transmission trees have proved successful in some situations (*Didelot et al., 2017*; *Jombart et al., 2014*; *Klinkenberg et al., 2017*; *Ypma, van Ballegooijen & Wallinga, 2013*). However, each of these make several assumptions, such as the presence of a complete transmission bottleneck that does not allow for repeated acquisition, a constant reproduction number across the outbreak, few or no unsampled cases, and that genetic variation is accumulated in pathogens in a clock-like fashion. The within-host diversity and horizontal transmission of mobile genetic elements in outbreak investigations complicates analyses further, and consequently, accurate reconstruction of exact transmission routes remains difficult for large and moderate-sized CPE outbreaks.

Our study has some limitations and points of note. Firstly, although most patient isolates had a clear epidemiological link to other isolates within the defined genomic clusters, not every isolate was able to be linked to another patient with KPC or a history of recent overseas travel, indicating a larger pool of unrecognised colonisation, though influenced in part by differing stringency in contact screening and data collection. For example, no patient in cluster A reported overseas travel since 1996, when KPC first emerged. Furthermore, although these patients were located at healthcare facilities in the same geographical region of Victoria and part of the same healthcare service, they did not have any overlapping inpatient admissions. Thus, it is likely that there are other individuals or environmental sources that serve as intermediary reservoirs of isolates not captured in our sampling, that facilitate ongoing transmission. Although environmental and contact screening was initially performed using methods published by others (*Centers for Disease Control and Prevention, US Department of Health and Human Services, 2015*), based on our data, the optimal approaches are still not well understood. Secondly, although the outbreak has slowed, cases are still emerging despite the extensive investigation and interventions implemented by the individual healthcare services. However, as shown in our analyses, many of these may be unrecognised colonisation following previous exposure rather than new transmission events. As a result of this knowledge, through the development of our guidelines, we have redirected infection control resources into intensive screening to identify individuals with asymptomatic colonisation, focussing efforts on healthcare facilities with ongoing transmission. Thirdly, as discussed above, capturing within-host population genomic diversity and reconstructing entire plasmid genomes from each clinical sample through sequencing would further enhance future outbreak investigations, but are currently limited by the cost of performing such analyses routinely. Sequencing cost and accuracy limitations also prohibit the routine use of more rapid sequencing technologies, such as PacBio SMRT and Oxford Nanopore sequencing, which could potentially provide more rapid results than Illumina sequencing to inform infection control decisions for prospectively sequenced isolates, though in the context of an outbreak spanning several years, turn-around times for Illumina sequencing can still be considered 'real-time'. Finally, although we conducted transmission inference analyses, these methods require further validation for routine implementation in outbreak investigations of multidrug-resistant pathogens. For example, we used *TransPhylo* in preference to other transmission modelling software as it did not require a high sampling proportion, and it provided the ability to account for and estimate the proportion of missing cases. However, it is difficult to know the true accuracy of these estimates, as even transmission chains arising from densely sampled wards were predicted to have unsampled cases by *TransPhylo*. Several possibilities to explain these unsampled cases include the limited sensitivity of the screening methods used, and the role of staff, visitors and the hospital environment in reconstructing transmission networks. Additionally, although transmission chains were predicted by *TransPhylo*, the resolution of the epidemiological data was insufficient to confirm the precise sequence of transmission events in a complex outbreak, with multiple potential pathways. Robust validation of these transmission models is required before being applied in outbreak

investigations to determine who infected whom—results that may subsequently have considerable ethical or legal implications.

## CONCLUSION

Our study reports a real-world prospective utilisation of WGS, including the difficulties encountered, to enhance a complex epidemiological investigation in real-time for an important pathogen. In contrast to most reported outbreaks of CPE within a single institution, our outbreak demanded a coordinated public health response. Previously the domain and responsibility of individual hospitals and healthcare facilities, the spread of CPE has become a public health issue. Through our experience, WGS has since been incorporated into our state guidelines for the management of CPE (*Department of Health & Human Services, Victoria State Government, 2015*). Developed during the outbreak investigation, this system entails routine WGS of all suspected CPE isolates with concurrent epidemiological investigation to allow prospective, centralised comparison of isolates and epidemiological data from multiple health services. In turn, this enables identification of potential transmission events between patients geographically and temporally dispersed at identification, translating into focussed interventions at the designated transmission locations. For such extensively drug-resistant organisms with limited treatment options, all feasible interventions towards reducing the early burden of disease are warranted.

## ACKNOWLEDGEMENTS

We wish to acknowledge and thank the referring clinical microbiology laboratories for facilitating transfer of isolates to MDU PHL, and the hospital infection control and infectious diseases staff who assisted and facilitated epidemiological data collection from the corresponding patients. We also thank Dr. Xavier Didelot, Imperial College London for his advice and assistance with *TransPhylo*, and Dr. Sally Partridge, University of Sydney for her expertise in Gram-negative plasmids.

### Funding

Funding for this study was provided by the Department of Health & Human Services, Victoria State Government and the Department of Microbiology and Immunology, University of Melbourne at the Peter Doherty Institute for Infection and Immunity. Jason C. Kwong (GNT1074824, and via the Centre for Research Excellence in Emerging Infectious Diseases, GNT1102962), Norelle L. Sherry (GNT1093468), Timothy P. Stinear (GNT1105525) and Benjamin P. Howden (GNT1105905) are supported by the National Health and Medical Research Council (NHMRC) of Australia. Sarah L. Baines is supported by an Australian Postgraduate Award. Jason C. Kwong, Sarah L. Baines and Norelle L. Sherry are also supported by Australian Government Research Training Program Scholarships. The funders had no role in study design, data collection and analysis, decision to publish, or preparation of the manuscript.

## Grant Disclosures

The following grant information was disclosed by the authors:

National Health and Medical Research Council (NHMRC) of Australia: GNT1074824, GNT1093468, GNT1105525, GNT1105905.

Centre for Research Excellence in Emerging Infectious Diseases (NHMRC): GNT1102962.

Australian Postgraduate Award.

Australian Government Research Training Program Scholarships.

## Competing Interests

Timothy P. Stinear is an Academic Editor for PeerJ.

## Author Contributions

- Jason C. Kwong conceived and designed the experiments, performed the experiments, analysed the data, contributed reagents/materials/analysis tools, wrote the paper, prepared figures and/or tables, reviewed drafts of the paper.
- Courtney R. Lane conceived and designed the experiments, performed the experiments, analysed the data, wrote the paper, prepared figures and/or tables, reviewed drafts of the paper.
- Finn Romanes conceived and designed the experiments, reviewed drafts of the paper.
- Anders Gonçalves da Silva performed the experiments, analysed the data, contributed reagents/materials/analysis tools, reviewed drafts of the paper.
- Marion Easton performed the experiments, reviewed drafts of the paper.
- Katie Cronin performed the experiments, reviewed drafts of the paper.
- Mary Jo Waters conceived and designed the experiments, reviewed drafts of the paper.
- Takehiro Tomita performed the experiments, reviewed drafts of the paper.
- Kerrie Stevens performed the experiments, reviewed drafts of the paper.
- Mark B. Schultz performed the experiments, reviewed drafts of the paper.
- Sarah L. Baines performed the experiments, reviewed drafts of the paper.
- Norelle L. Sherry performed the experiments, reviewed drafts of the paper.
- Glen P. Carter performed the experiments, reviewed drafts of the paper.
- Andre Mu performed the experiments, reviewed drafts of the paper.
- Michelle Sait performed the experiments, reviewed drafts of the paper.
- Susan A. Ballard performed the experiments, reviewed drafts of the paper.
- Torsten Seemann conceived and designed the experiments, analysed the data, contributed reagents/materials/analysis tools, reviewed drafts of the paper.
- Timothy P. Stinear conceived and designed the experiments, analysed the data, wrote the paper, reviewed drafts of the paper.
- Benjamin P. Howden conceived and designed the experiments, analysed the data, wrote the paper, reviewed drafts of the paper.

## Human Ethics

The following information was supplied relating to ethical approvals (i.e. approving body and any reference numbers):

Data were collected as part of an outbreak investigation through the Victorian Department of Health and Human Services under the Public Health and Wellbeing Act 2008 (https://www2.health.vic.gov.au/about/legislation/public-health-and-wellbeing-act).

## DNA Deposition

The following information was supplied regarding the deposition of DNA sequences:

The raw Illumina short-read sequencing data and completed PacBio genome assemblies supporting the conclusions of this article are available in the National Center for Biotechnology Information database under BioProject PRJNA397262.

## Data Availability

In-house scripts used in the bioinformatics analyses are available at https://github.com/tseemann and https://github.com/kwongj.

## Supplemental Information

Supplemental information for this article can be found online at http://dx.doi.org/10.7717/peerj.4210#supplemental-information.

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
