# Peer review of "Translating genomics into practice for real-time surveillance and response to carbapenemase-producing Enterobacteriaceae: evidence from a complex multi-institutional KPC outbreak"

_PeerJ, doi:10.7717/peerj.4210_

## Round 0.1 · original submission · Minor Revisions

Thank you for submitting this very interesting article to PeerJ. As you can see both reviewers are favourably disposed to publication. They have suggested some minor changes to the manuscript and I am happy for you to make any changes as you see fit.

·

Basic reporting

No comment

Experimental design

No comment.

Validity of the findings

No comment.

Additional comments

This article describes the enhanced effect of combining WGS information with traditional field epidemiology to inform management of an established KPC producing Enterobacteriaceae outbreak.The text is well written, clear and unambiguous with sufficient relevant references and background provided.

General comments

The value of the WGS in the context of patient benefit and healthcare economics is difficult to quantify and the manuscript would benefit from estimates of these in this study, i.e. the authors state that further transmission has slowed because of increased intervention but is there an estimate of the healthcare saving and patient benefit because of this? This is a subjective area but if screening had been introduced at an earlier stage would there be an estimated saving in terms of hospital bed days for example?

The authors discuss the revision of guidelines for outbreak management since the study was performed, a discussion of further improvements/recommendations and how they would be implemented would be of value.

Throughout the manuscript the term real-time is used to describe how WGS informed transmission dynamics and subsequent infection control management. Near real-time is more accurate as it still takes at least a week to take appropriate samples, culture, sequence and analyse the information.

The ability to determine whom infected whom in these investigations is an interesting point for discussion and also has potentially difficult ethical/legal considerations that could also be discussed.

·

Basic reporting

The article is clearly written and easy to follow, with an extensive bibliography and appropriate context provided via the Introduction.

DNA Data Checks:
The raw data and assemblies, along with a useful assortment of sample-level metadata, have been deposited at NCBI and I was able to successfully access them. The accession number is included in the manuscript. Scripts used in the analyses have also been made available via two GitHub repos.

Human Participant Checks:
The investigation was undertaken by a public health agency as part of an outbreak response. No identifying information is included in the article or in the publicly available data.

Experimental design

The article is well-suited to publication in PeerJ, offering an extremely comprehensive investigation of a pathogen of international concern. The objective of the study is clearly described, and the work presents an exemplary model for uniting genomic and epidemiological investigations of hospital-associated infections. Each of the many analyses was done to high technical standards, and described in enough detail for others to be able to replicate the work.

Validity of the findings

The conclusions around the origins and the transmission of the outbreak isolates are well-supported by the data, and clearly linked to downstream infection control activities. Limitations of the study are clearly identified.

Additional comments

This is wonderfully detailed and fascinating investigation of CPEs circulating in Victoria, Australia over 2012-2015, which links genomic and epidemiological data in a way that very few papers in this domain have. It was a very enjoyable read, and my suggestions for improvement are all very minor and addressing them is left to the author's discretion.

Introduction
1. The abstract mentions that the genomic epidemiology investigation proved more fruitful than genomics or traditional epidemiology alone. Perhaps towards the end of the Introduction, the authors could describe what the state of the epidemiological understanding was at the time the genomic investigation was launched - were there obvious gaps that the genomic analysis was aimed at closing?

Methods
1. The authors are to be commended for the incredible breadth and depth of their analyses! That being said, there is definitely a lot for the reader to keep track of here, so perhaps a figure outlining the study workflow could be provided to orient the reader.

2. Along those lines, the Bioinformatics subsection contains many analyses - these could be further subdivided into smaller sections to facilitate reading.

Results
1. Line 310: Would be helpful to briefly differentiate KPC-2 and KPC-3 earlier in the manuscript for the reader unfamiliar with these designations - perhaps a brief comment in the Introduction, including their incidence in other settings like the US.

2. Line 311: "from one patient" instead of "from the one patient"

3. Figure 1: The annual peaks in Sept/Oct/Nov are interesting and might merit some comment in the Conclusions section.

4. Line 313: delete the "of"

5. Line 315: For the readers who skip the Intro and Methods and head straight to result, it would be helpful to explain what the "Initial Analysis" was - from my reading of the methods, it was a retrospective 2014 look at the isolates from the 29 patients identified to date, with further work happening prospectively.

6. Figure 2: Depending on the actual geography, it might be interesting to colour the facility nodes according to their distance from each other, maybe with the most central Melbourne facility as a saturated colour with the other facilities less saturated as one moves out. This would, of course, only be really interesting if there was some spatial structure to the phylogeny, but it is suggested in the Combined Analysis section. It would be interesting to note Facility F, though, which is mentioned later on in the text.

7. Figure 4: No suggestions here, just wanted to compliment you on this super figure!

8. Line 392: A lot of the TransPhylo transmission events arise from unsampled individuals (white nodes), whereas I imagine the screening in place knowing that KPC was circulating meant that most of the population should have been fully sampled. I think (as a TransPhylo author ;) that this is a bit of a shortcoming of this current implementation – I think it sometimes overestimates the proportion of unsampled cases. This might be worth commenting on, as would noting whether any of the chains it did identify amongst the sequenced isolates (I spot an interesting one involving five B1-harbouring individuals) were reflected in the epidemiology.

9. Line 430: I wonder if a figure summarizing the plasmid content of each isolate might be helpful here, to give a quick visual of which isolates across which species harboured identical or near-identical plasmids.

10. Line 475: What sort of accessory genome changes?

11. Figure 9: I see environmental sampling isolates in this tree that appear to be closely linked to one of the patients for whom multiple isolates were sequenced (bright pink patient) and another singly-sequenced patient. This is interesting, but I don’t see a mention of it in the main text.

Supplement:
1. Figure S6: The black (overall) regression line seems to be missing, though the R^2 value is there.

---

## Round 0.2 · accepted · Accept

Thank you for your considered responses to the reviewer comments and improved manuscript which I am delighted to accept for publication.